# Retinol and α-Tocopherol Contents, Fat Color, and Lipid Oxidation as Traceability Tools of the Feeding System in Suckling Payoya Kids

**DOI:** 10.3390/ani12010104

**Published:** 2022-01-02

**Authors:** Mercedes Roncero-Díaz, Begoña Panea, María de Guía Córdoba, Anastasio Argüello, María J. Alcalde

**Affiliations:** 1Department of Agronomy, Universidad de Sevilla, Ctra. Utrera km. 1, 41013 Seville, Spain; mroncerodiaz@gmail.com; 2Unidad de Producción y Sanidad Animal, Centro de Investigación y Tecnología Agroalimentaria de Aragón (CITA), Avda. Montañana 930, 50059 Zaragoza, Spain; bpanea@cita-aragon.es; 3Instituto Agroalimentario de Aragón—IA2 (CITA-Universidad de Zaragoza), C/Miguel Servet, 50059 Zaragoza, Spain; 4Department of Animal Production and Food Science, University of Extremadura, Av. Adolfo Suarez, s/n, 06007 Badajoz, Spain; mdeguia@unex.es; 5Animal Production and Biotechnology Group, Institute of Animal Health and Food Safety, Universidad de Las Palmas de Gran Canaria, 35413 Arucas, Spain; tacho@ulpgc.es

**Keywords:** goat kid, fat-soluble vitamins, color, traceability, lipid oxidation, feeding systems

## Abstract

**Simple Summary:**

In Spain, goat farms are mainly oriented to milk production, although kid meat contributes to their sustainability, particularly in autochthonous breeds such as Payoya. Usually, kids are fed artificial milk until slaughter, allowing the use of goat milk for the commercialization of cheese, but several studies indicate that feeding kids natural milk improve the quality of their meat. The aim of the present study was to find traceability markers to discriminate between kids that are fed natural milk (with different goat management systems) and those fed a milk replacer. For this purpose, we proposed the quantification of retinol and α-tocopherol contents in plasma and fat, the amount of kidney fat, lipid oxidation, and some fat color parameters as potential markers. The results showed that plasma retinol concentrations were higher in kids fed feeding systems with synthetic vitamins. The plasma α-tocopherol concentrations were higher in kids fed grass-based feeding systems (which contain the natural forms of these vitamins). A dilution effect was shown for the retinol concentration in fat. Collectively, the analyzed variables allowed a discriminant analysis to correctly classify kids according to their feeding system and could ensure traceability to consumers.

**Abstract:**

The effects of Payoya kid feeding systems on the fat-soluble vitamin (retinol/α-tocopherol) contents, fat content, fat color, and the oxidation index were evaluated to determine their potential for use as feeding system traceability tools. Four groups of Payoya kids (55 animals in total) fed milk exclusively were studied: a group fed a milk replacer (MR) and three groups fed natural milk from dams reared with different management systems (mountain grazing (MG), cultivated meadow (CM) and total mixed ration (TMR)). Kids were slaughtered around one month of age and 8 kg of live weight. Kids from the MG and CM groups presented lower retinol (5.56 and 3.72 µg/mL) and higher α-tocopherol plasma (11.43 and 8.85 µg/mL) concentrations than those from the TMR and MR groups (14.98 and 22.47 µg/mL of retinol; 2.49 and 0.52 µg/mL of α-tocopherol, respectively) (*p* < 0.001). With respect to fat, kids with a higher intramuscular fat percentage (CM and TMR groups) had lower retinol contents (16.52 and 15.99 µg/mL, respectively) than kids from the MG and MR groups (26.81 and 22.63 µg/mL, respectively) (*p* < 0.001). A dilution effect of vitamins on fat was shown: the higher the amount of fat, the lower the vitamin concentrations, the higher the lipid oxidation index (MDA), and the lower the SUM (absolute value of the integral of the translated spectra between 450 and 510 nm). A discriminant analysis that included all studied variables showed that 94.4% of the kids were classified correctly according to their feeding system and could allow traceability to the consumer.

## 1. Introduction

Goat kid meat has a low content of fat and high contents of branched-chain fatty acids compared to other traditional meats and, therefore, can be beneficial to human health [1]. In Spain, 75% of goat production is aimed at dairy production [2]. According to official data (MAPA, 2019) [3], 91% of this milk is transformed into cheese. Southern Spain (Andalusia) produces 42.1% of the Spanish goat’s milk, so 220,000 t are transformed into cheese. In addition, 36.8% of the total national production of goat meat is derived from Andalusia, making it the region with the highest quantity of slaughtered suckling kids (32.34%).

The Payoya goat breed is mainly located in the Andalusian regions of the Sierra de Cádiz and the Serranía de Ronda (Málaga). It is one of the areas with the longest tradition of dairy goat farming. The production of quality traditional cheeses based on Payoya goat milk is an important economic factor in these regions. Meat production from this breed is much lower than dairy production, although it is of excellent quality and represents only 18.5% of revenue [4].

Goats can be reared in intensive or semi-extensive conditions. In the former, the goats are housed and are fed a total mixed ratio with vitamin supplementation. In the latter, the feeding of the goats is based on grazing, although with some supplementation with compound feed. On their own, suckling kids are typically fed under one of the two following feeding systems: with a milk replacer [5], which allows the commercialization of their dams’ milk, or with natural milk, which, according to some authors [6], improves kid meat quality.

Additionally, food products of differentiated quality are important tools for improving marketing; thus, one of the main strengths of goat farming based on the Payoya breed is the importance of grazing as a basis for feeding goats and is therefore linked to sustainable activities [7]. The value of sustainability is rising, especially in its environmental aspects, which large food chains have begun to use in advertising since it is highly valued by consumers. Thus, strategic actions are necessary to enhance farms’ viability and profitability [8], including reasonable income levels for the farmers.

With respect to purchasing meat, in addition to intrinsic quality cues, such as quality, consumers are increasingly interested in extrinsic cues, such as animal welfare, local production, the methodologies used in food production [2], production systems [9] healthy properties [10] and quality marks and information in labels [11]. Meat consumption, in recent years, does not have suitable publicity. Studies on traceability are an important tool to contribute to the change of that image and promote their frequency of consumption. Thus, knowing how kids were reared is important to ethically minded consumers, especially through data regarding the feeding systems used and traceability tools [12,13].

The carotenoids and vitamins A and E cannot be synthesized by mammals, so they must be added to animal diets [14,15]. Therefore, these compounds may be suitable candidates for traceability biomarkers [14,15]. It has been documented in goats that these compounds, when present in animal feed, can appear in the plasma, milk, and tissues of the animal [16]. However, for kids, there is scarce literature regarding whether these compounds can pass from goat feed to their milk and from milk to the suckling kid’s plasma or tissues.

Vitamins A and E have antioxidant properties [17] and can protect lipids from oxidation. Therefore, the susceptibility of fat to oxidation can be used as an indirect biomarker of vitamin content. Unfortunately, both vitamin quantification and lipid oxidation determinations are expensive and time-consuming.

Consequently, non-destructive analysis (e.g., fat color analysis) could be very useful for traceability purposes. The literature states that grass-based feeding regimes produce yellower fats, so fat color may be able to be used as an indirect marker of an animal’s feeding system [18]. Calculation of the absolute value of the integral of the translated spectra of fat tissue reflectance between 450 and 510 nm [19] (the region where carotenoids absorb light associated with the trichromatic coordinates of the CIELAB space) has been proposed as an indicator of meat from grass-fed animals [20,21].

Álvarez et al. (2014) [20] used the concentrations of carotenoids, vitamin A, and vitamin E in plasma and adipose tissue, as well as fat color, as traceability parameters to distinguish animals that grazed from those fed compound diets. More recent studies carried out in lambs [22,23] suggested that maternal feeding can be traced by quantifying the carotenoid and tocopherol contents of meat, although this tracing was hampered when mixed diets or different indoor fattening systems were used in lambs after weaning. On the other hand, Ripoll et al. [5] studied the color of meat and kidney fat as functions of the feeding system in two goat breeds (Malagueña-Murciano Granadina) and concluded that the breed had a greater influence on color than the type of lactation (natural milk vs. a milk replacer). This lack of agreement implies that this subject is far from clear. However, this subject has scarcely been studied on kids [5]. Therefore, the objectives of the present study were (1) to investigate whether carotenoids and vitamins can pass from the maternal milk of goats reared under three different regimes (mountain grazing, cultivated meadow, and total mixed diet) to the suckling kids’ plasma and fat and (2) to determine whether vitamin contents in the plasma and fat, as well as fat oxidation and color, can be used to determine whether the kids were fed with natural milk or with a milk replacer, that is, whether these parameters can be used as traceability tools.

## 2. Materials and Methods

### 2.1. Animal Handling and Feeding

All animal management procedures were conducted according to the guidelines of Directive 2010/63/EU on the protection of animals used for experimental and other scientific purposes [24].

This study was conducted in Sierra de Grazalema (Cádiz), in Southern Spain, with an average altitude of 800 m. This area is characterized by a semiarid Mediterranean climate, with an average annual precipitation of 2093.92 mm/m^2^/year [25], with seasonal distribution from October to April, wet and cold winters, and dry, hot summers. This climate offers high- and low-production grazing periods, with high production periods in autumn and spring and low production usually in summer [26]. Goat farms were located in the municipality of Grazalema (36°47′56.43″ N, 5°19′57.91″ W and 36°44′45″ N, 5°24′21″ W) and in the municipality of El Bosque (36°43′47″ N, 5°30′47″ W). The experimental period was during the spring.

We used a total of 55 suckling kids from the Payoya breed [27], all of which were born within 2–3 days, males, healthy, from a single birth, and in natural birth. During the trial duration, kids were fed only milk *ad libitum*, with no dietetic complement. In all cases, the kids remained stable. The animals were distributed across the four following feeding systems:Mountain grazing (MG): 12 suckling kids were slaughtered at an average age of 37 days. During lactation, their mothers grazed in a dehesa system [28] for 8 h and were stabled for the remaining time. In addition, dams were supplemented with 800 g of commercial feed.Cultivated meadow (CM): 13 suckling kids were slaughtered at an average age of 32 days. During lactation, their mothers grazed in a cultivated pasture of oats for 8 h (the remainder of the time, they were stabled) and were supplemented with 500 g of compound feed (28.45% of the same commercial feed as the MG kid + 35% sunflower seed (*Helianthus annuus*) + 28.3% oat seed (*Avena sativa*) + 8.25% pea seed (*Pisum sativum*)).Total mixed ration (TMR): 14 suckling kids were slaughtered at an average age of 30 days. Their mothers were kept permanently indoors and fed 1.5 kg of the same commercial feed and hay *ad libitum.*Further details of the dam feeding systems and management were described in a previous study [29].Milk replacer (MR): Sixteen kids were fed a milk replacer (CORDEVIT calostrado-50, Lemasa, Spain) enhanced with 80,000 IU/kg of vitamin A and 30 IU/kg of vitamin E. The composition of the milk replacer administered to kids is shown in Table 1. They were slaughtered at 47 days old. These kids ingested approximately 1.25 L of milk per day [30], with a concentration of 200 g of power per liter, as indicated in the product instructions. Table 1 shows the composition of the milk replacer.

### 2.2. Slaughter, Plasma Collection, Carcass Characteristics, and Sampling

Kids were slaughtered when they reached a live weight of 8 kg. They were weighed when they arrived at the slaughterhouse (slaughter live weight, SLW). The slaughter was carried out in accordance with the European regulation [31] after 12 h of fasting with free access to water. In the exsanguination, blood samples were taken from the jugular vein in vacuum tubes (VACUTAINERTM; Becton, Dickison and company, Franklin Lakes, NJ, USA) using Li-heparin as an anticoagulant. After transporting blood samples to the laboratory at 4 °C, they were centrifuged (2500 rpm, 10 min, 4 °C; Eppendorf 5810 R Eppendorf, Hamburg, Germany). Plasma was frozen at −20 °C until analysis.

Carcasses were weighed (hot carcass weight, HCW) and cooled for 24 h to 4 °C in the dark. Subsequently, the kidney fat was extracted and weighed (kidney fat weight, KFW) with a bascule Nahita Blue Series S162, and kidney fat color was assessed at that time. Then, the *Longissimus thoracis* muscle, between the 1st and 13th ribs, and intermuscular fat (between this muscle and those adjacent to it) were extracted from the left side of the carcass, vacuum packed, and frozen at −20 °C until analysis. Intermuscular fat was used to quantify fat-soluble vitamins, whereas the *Longissimus thoracis* was used to determine the intramuscular fat content (LTF) and lipid oxidation index (MDA).

### 2.3. Analytical Procedures

#### 2.3.1. Fat Color

Kidney fat color values were recorded at 24 h post mortem from three locations, randomly selected but avoiding blood spots, discolorations, and less covered areas. Fat color was measured using a Minolta CM-700d spectrophotometer (Konica Minolta Holdings, Inc., Osaka, Japan) in the CIELAB space [32], with a measured area diameter of 8 mm, a specular component included and 0% UV, and standard illuminant D65, which simulates daylight (color temperature 6504 K), an observer angle of 10° and zero and white calibration. Lightness (L*), redness (a*), and yellowness (b*) were recorded. Hue (h°) and chroma or saturation (C*) were calculated as h° = tan^−1^(b*/a*) (expressed in degrees) and C* = $\sqrt{{\left(a*\right)}^{2}+{\left(b*\right)}^{2}}$.

Reflectance spectra in the visible region between 450 and 510 nm (with 10 nm increments) were also acquired and recorded to obtain the translated reflectance value (TRi) and to estimate the absolute value of the integral of these data (SUM). In previous studies [20,21,33], the reflectance spectra between 510 and 450 nm were translated to make the reflectance value at 510 nm equal to zero (TR). The TRi was calculated from the reflectance value (Ri) as follows: TRi = Ri-R510, with *i*= 450, 460,… 510; whereas the SUM was calculated according to the following formula:SUM = [(TR_450_/2) + TR_460_ + TR_470_ + TR_480_ + TR_490_ + TR_500_ + (TR_510_/2)] × 10

An extensive explanation of the baselines of the method is exposed in Prache and Theriez (1999) [19].

#### 2.3.2. Intramuscular Fat Amount, Lipid Oxidation, and Vitamin Extraction

Intramuscular fat was extracted following the procedure described by Bligh and Dyer (1959) [34]. In total, 15 g of meat was treated with 45 mL of a 1:2 chloroform-methanol solution, homogenized for two minutes, and then centrifuged for 5 min at 6000 rpm. The supernatant was collected in a new centrifuge tube. The sediment was washed with 15 mL of chloroform, homogenized for 30 s, and centrifuged as before. The supernatant was collected again, after which 15 mL of 0.8% NaCl was added. This mixture was centrifuged again for 5 min at 6000 rpm. After centrifugation, two phases were observed: the chloroform phase and the methanolic phase. With a Pasteur pipette, the methanolic phase was removed. To remove traces of water, the extract was filtered with anhydrous sodium sulfate in a flask of known weight, the solvent was evaporated in a rotary evaporator at a temperature not exceeding 43 °C, and the weight of the fat was calculated by the weight difference.

Lipid oxidation was measured using the TBARS method described by Sørensen and Jørgensen (1996) [35], and the result is expressed as mg malonaldehyde (MDA) kg^−1^ of meat. Starting with a 2.5–3 g of sample of meat, 9 mL of trichloroacetic acid (TCA) solution was added (TCA was composed of 1 g of propylgalate, 1 g of ethylenediaminetetraacetic acid (EDTA), and 75 g of TCA per liter of extra pure water) and homogenized for 45 s. Then, the sample was centrifuged for 5 min at 10,000 rpm. The supernatant was removed, and the acidic extract was collected. A total of 3 mL of TBA (288.3 mg per 100 mL of ultra-pure water) was added to each tube. The tubes were placed in a 100 °C water bath for 40 min. Subsequently, the samples were cooled in a thread bath (10 min) and centrifuged at 4500 rpm for 10 min. Finally, the absorbance was measured on a spectrophotometer at two wavelengths: 532 and 600 nm. The quantification of TBARS was carried out through a calibration line with known solutions of malondialdehyde (MDA).

The extraction of retinol and α-tocopherol was carried out following published methodology by Nozière et al. (2006) [36]. Intermuscular fat (500 mg) was mixed with 1 mL of 3,5-di-tert-4-butylhydroxytoluene (BHT) ethanolic solution (12%, *w*/*v*), after which 2.5 mL of sodium hydroxide in ethanol 30% (*w*/*v*) and 5 mL of ethanol were added. The saponification reaction was carried out overnight at room temperature and in the dark. Water was added to stop the reaction, and the analytes were extracted with 10 mL of an ether/hexane mixture (2:1, *v*/*v*). The extracted samples were centrifuged (3500 rpm, 10 min, 4 °C) (Eppendorf 5810 R centrifuge), and the upper organic phase was collected. The extraction was repeated twice. The organic phases were pooled, washed several times with water, collected, and dried (using nitrogen gas). The residue was dissolved in 1.5 mL of ethyl acetate and filtered for HPLC analysis under the same conditions explained for plasma.

#### 2.3.3. Extraction and Determinations of Vitamins from Plasma

Vitamin extraction from plasma was performed in duplicate using the protocol described by Lyan et al. (2001) [37]. For each extraction, 2 mL of plasma was diluted with 1 mL of distilled water. Subsequently, 2 mL of ethanol and 2 mL of hexane were added. The organic phase was separated with Pasteur pipettes after centrifugation (2500 rpm, 10 min, 4 °C). During the aqueous phase, hexane extraction was repeated twice. The extracts were evaporated to dryness with a stream of nitrogen gas. The dry residue was resuspended with 50 μL of ethyl acetate for subsequent analysis by HPLC.

### 2.4. HPLC Conditions

HPLC analysis of the samples was performed on a Varian Pro STAR 240 system equipped with a photodiode detector, a quaternary pump, a temperature control module set at 20 °C, an automatic injector (injection volume of 20 μL), and a Hypersil ODS C18 column (150 × 4.6 mm, 5 μm). The elusion gradient described by Mouly et al. [38] was used, with some modifications [39]. The mobile phase was composed of methanol (MeOH), methyl-tert-butyl-ether (MTBE) and water added at different proportions over the measurement time according to the following gradient: 0 min: 90% MeOH + 5% MTBE + 5% water; 12 min: 95% MeOH + 5% MTBE; 25 min: 89% MeOH + 11% MTBE; 40 min: 75% MeOH + 25% MTBE; 50 min: 40% MeOH + 60% MTBE; 56 min: 15% MeOH + 85% MTBE; 62 min: 90% MeOH + 5% MTBE + 5% water. The mobile phase was pumped at a rate of 1 mL/min, and the chromatograms were monitored at 450 nm for the carotenoids, 325 nm for retinol, and 280 nm for α-tocopherol. Every day at the end of the analysis, the column was washed with MTBE:MeOH (50:50) for 20 min. The chromatograms were monitored at 450 nm for carotenoids, at 325 nm for retinol, and at 280 nm for α-tocopherol. The compounds were identified using standards from Sigma Chemical (Madrid, Spain). Quantification was performed using standard solution calibration curves. Standards of the carotenoids and fat-soluble vitamins with a degree of purity >90% were acquired from Aldrich-Fluka-Sigma Chemical (all-trans-retinol and α-tocopherol).

### 2.5. Statistical Analysis

For the statistical analysis, IBM SPSS Statistics 25.0 software for Windows (March 2017) was used. The statistical model used in the analysis of variance (ANOVA) included the fixed effect of the kid feeding system and the random effect of the individual. The model used for each parameter was: Yijk = μ+ PSi + εijk. Where Yijk = observations for dependent variables; μ = overall mean; PSi = fixed effect of feeding system ((i = MG, CM, TMR or MR), and εijk = random effect of residual. Significant differences between means were determined by a post-hoc Tukey test.

Finally, three discriminant analyses (DAs) were carried out using a stepwise model considering the feeding systems of the animals as the classification factor. The discriminant classification method was leave-one-out cross-validation. The first DA was performed with carcass parameters, kidney fat color parameters, and the lipid oxidation index. The second DA was based on the levels of vitamins in the kids’ deposits. The third DA was based on all variables from the previous discriminant analyses.

## 3. Results and Discussion

### 3.1. Carcass Traits and Kidney Fat Color Parameters

The results of the variables measured at slaughter and the color parameters of kidney fat are shown in Table 2. The criterion for slaughter was a live weight of 8 kg; no significant differences were found between the four feeding systems. However, the KFW and the HCW variables differed according to the feeding system. CM kids presented the lowest HCW values (*p* = 0.002) and the highest KFW values (*p* < 0.001). MR kids took the longest time to reach the target weight, being 47 days old on average, while TMR kids took only 30 days.

Goats are “white fat” animals [40]; thus, the intestinal conversion of β-carotene to retinol is very efficient, and almost nothing is absorbed as β-carotene [41]. The young age of the animals limited fat deposition and the cumulative effect of carotenoids and fat-soluble vitamins [33]. However, the color parameters of kidney fat were studied because other compounds, such as chlorophyll, in the maternal diet affect the fat color [42] in addition to carotenoids.

All the fat color variables except for L* and h° were affected by the feeding system. Kids were slaughtered very young at a similar age, so the fat lightness measures were very high and similar for all groups. Fat from CM kids presented the lowest values of redness (a*), yellowness (b*), and saturation (C*). MR kids were the oldest animals at slaughter, maybe allowing a greater amount of pigments to accumulate in the fat [43]. It was also observed that CM kids exhibited lower b* values (*p* = 0.037), lower b* values (*p* = 0.037), and lower C* values (*p* = 0.035) than MG kids, which could be partially explained by a dilution effect of the pigments into the fat [22,44] since CM kids presented the highest amount of kidney fat.

To the best of our knowledge, there are studies on the fat color in animals with different management systems, but there are no studies on these parameters as traceability markers in the management of maternal feeding in suckling kids. Current values for kidney fat color variables are similar to those reported by Ripoll et al. [5], who studied kids from the Murciano-Granadina and Malagueña breeds with ages and weights similar to ours. The effect of the feeding regime on fat color in lambs has been described as a tool to trace by other authors [21]. In this way, Ripoll et al. [45] indicated that the feeding regime had an effect on color variables, with higher L*, b*, and C* values in fat from grazing system-fed lambs than in fat from indoor-fed lambs. Moreover, Álvarez et al. [20] concluded that light lambs fed from grazing systems had higher values of L*, a*, b*, and C* than animals fed diets based on compound feed, although these differences were not significant. Most of the studies showed differences between the color parameters because the feeding regimes of the animals were very different. The difference between these studies in relation to our study is that in the current study, the different management systems analyzed were those of the dams, and the fat color parameters were analyzed in the kids. Ripoll et al. [5] showed that breed affected the fat color more than the rearing system, but our results showed that the most important factor that affects fat color is the degree of fatness and the cumulative effect of the pigments with age rather than the breed or the rearing system.

Our study detected an effect of the feeding system on the SUM (Table 2). Regarding animals fed natural milk, MG and CM kids (whose dams were fed diet grass-based) presented higher values than TMR kids (whose dams were fed a diet based on compound feed), although significant differences (*p* = 0.007) were observed only between MG and TMR kids. No significant differences were found between animals fed natural milk (CM and MG), and animals fed a milk replacer (MR), which was in agreement with the results of Ripoll et al. [5,46]. The SUM is an index related to the carotenoid content in fat [33], and because of this, it has been used as a food traceability biomarker in lambs [20,33,45,47,48] to differentiate between management systems (grazing vs. concentrate) [20,21]. The results from the current study agree in part with those from previous studies [20,21] because animals from grass-based feeding regimes had higher SUM values than animals from concentrate-based feeding regimes. The SUM values of the TMR kids were similar to those obtained in studies carried out in indoor lambs that directly received compound feed as the whole feeding system [20] or as a complement to a dairy diet [21,47]. However, the SUM values of the current experiment are, in general, lower than those found in the literature, because our kids were fed only goat milk, while the animals in other studies ingested grass directly and could also be influenced by other factors, such as the low transfer of carotenoids from the dams’ feeding to milk [36], low transfer from milk to the kids’ tissues [49], low deposition in the perirenal fat due to the young age of these animals [33], or differences in fat deposition patterns between lambs and kids [10].

The reflectance spectra patterns are presented in Figure 1. The % TR between 450 and 510 nm showed a different pattern for TMR kids than for the other kids. TMR kids had positive % TR values above the abscissa axis (+0.79) for 490 nm, while those for the other feeding systems were negative: MG (−0.54), CM (−0.42), and MR (−0.39). Nevertheless, at 500 nm, the four groups presented positive % TR values, and there were no differences between MG (+0.25), CM (+0.22), and AL (+0.26) kids, whereas TMR kids still presented the highest value (+0.97).

The absolute value of the integral (SUM) is defined as the area that is comprised between the curve and X-axis in the zone 450–510 nm. In agreement with the results presented in Figure 1 and Table 2, MG kids registered a greater area than TMR kids, which agrees with the results from other authors [33]. This appears to be the first study that compares the SUM and the %TR between 450 and 510 nm in the fat of kids fed natural milk (with different goat management systems) and a milk replacer.

### 3.2. Intramuscular Fat Content and Lipid Oxidation

Regarding the percentage of intramuscular fat, significant differences were found (*p* < 0.001) between the four feeding systems (Table 3). CM and TMR kids showed a similar content, and the values were three or four times higher than those of MG and MR kids. The values from MG and MR kids were similar to those indicated by other authors in the Payoya breed [50] and other Spanish goat breeds [1], as pointed out by the authors of [51] where intramuscular fat is around 2% in such young kids; however, the CM and TMR groups were well above this reference value. CM and TMR kids reached the slaughter weight earlier; that is, they presented higher growth rates than the other kids (Table 2), which could indicate that they received milk with a higher energy content (maybe because of a higher quality of feed received by their dams). It is known that when a goat's diet is supplemented with seeds rich in linoleic and/or linolenic acid, there is an increase in fat deposition [52]. In this sense, dams from CM kids received in their diet sunflower pulp, oat seeds, and peas, whose linoleic and/or linolenic acid fatty acid content ranges from 35% to 65% [53,54,55], and dams from TMR kids were fed a compound feed with an oil and crude fat content of 4.14%. These factors could explain the higher fat percentage in CM and TMR kids than in MG and Al kids.

The lipid oxidation index (MDA) was affected by the feeding system (Table 3). CM and TMR kids showed significantly (*p* < 0.001) higher values than MG and MR kids. The greater the amount of fat, the greater the lipid oxidation index, as expected. It is known that fat-soluble compounds (retinol, α-tocopherol, and carotenoids) can act as antioxidants [56], inhibiting the oxidation of lipids in meat. Therefore, the lower amount of retinol in the fat of the CM and TMR kids could have influenced these results.

Some authors [52] stated that the fat composition of goat meat can be modified by supplementing the diet with polyunsaturated fatty acids. In addition, the feeding regime of the goats determines the fat content of the milk that the kids consume [26,52], modifying, in turn, the fat profile of the kids’ meat [57] since they do not hydrogenate them in the rumen. Thus, it is logical to believe that dams from CM and TMR kids, with diets rich in linoleic acid and linolenic acid, have milk with a higher content of polyunsaturated fatty acids that would pass to the adipose tissue [26,58] of their kids, making their fat more susceptible to oxidation [59] and therefore with higher MDA values.

### 3.3. Retinol and α-Tocopherol Contents

#### 3.3.1. Concentrations in Intermuscular Fat

Retinol was detected in the fat of the animals from the four feeding systems (Table 3). Statistically significant differences (*p* < 0.001) were found between the feeding systems, the concentrations being similar two to two, with retinol concentrations in the MG and MR kids similar to each other and significantly higher than the concentrations in the TMR and CM kids.

Álvarez et al. [20] analyzed the amount of retinol in lamb kidney fat, and Blanco et al. [22] analyzed the concentrations of retinol in the kidney and subcutaneous fat. Both studies found that fat from grass-fed lambs contained more retinol than fat from compound-fed lambs. Rufino-Moya et al. [23] obtained the same results as previous authors [20,22] regarding kidney and subcutaneous fat while observing no differences in intramuscular fat. The fat retinol values in our study were higher than those obtained by Blanco et al. [22] in the kidneys and subcutaneous fat of suckling lambs and by Rufino-Moya et al. [23] in intramuscular fat, kidneys, and subcutaneous fat of suckling lambs that were fattened with dietary concentrates after weaning. On the other hand, Osorio et al. [60] stated that kids that were fed a milk replacer had higher retinol concentrations in the Longissimus dorsi muscle and in various muscles of the leg than kids that were fed natural milk. The current results show that the amount of fat retinol is related to the amount of fat deposited in the tissues, with a consequent dilution effect [44], which is in agreement with other studies that observed that the higher the fat content, the lower retinol content in lambs supplemented with vitamin A [44,61]. The distribution of retinol in target tissues occurs through retinol transporter protein type 4 (RBP4) [62]. All the physiological functions of this protein and the factors that influence these functions are not yet well understood, but they seem to be related to an increase in subcutaneous fat with age and an increase in intramuscular fat in cattle [63]. Therefore, the young age and low amount of fat exhibited by the animals in the current experiment would explain the fact that we did not find a clear explanation for the retinol deposit in the intermuscular fat of goat kids.

α-Tocopherol was not detected in intermuscular fat regardless of the feeding system used. According to Debier and Larondelle [64], α-tocopherol storage differs depending on the type of tissue, being very slow to appear in adipose tissue, even slower than retinol. The animals used in our experiment were very young when they were slaughtered. Studies conducted in suckling [20,22,23,60] and in light lambs with different types of feeding regimes [20,61,65] reported tocopherol in fat, but all the animals were older than those used in the present study.

#### 3.3.2. Concentration in Plasma

The feeding system affected both the retinol (*p* ˂ 0.001) and the α-tocopherol (*p* ˂ 0.001) plasma contents (Table 3). The kids whose dams were fed grass-based diets (MG and CM) presented lower retinol and higher α-tocopherol concentrations than the kids whose dams received synthetic vitamins (TMR), the same pattern observed in their dams [29]. In addition, MR kids registered higher plasma retinol levels (*p* ˂ 0.001) than kids fed natural milk (TMR, MG, and CM), which agrees with the results reported by other authors [60]. Obviously, vitamin A supplementation in MR kids, provided directly by the milk replacer, was greater than the vitamin intake in TMR kids, in which vitamins come from dams’ milk, with possible losses of vitamins resulting from ruminal action and intestinal absorption [41,66] in goats. On the other hand, as could be seen in a previous experiment of the series [29], TMR goats showed higher plasma retinol concentrations than grass-fed goats (MG and CM) and, therefore, higher retinol concentrations in their milk. It must be taken into account that the vitamin supply to TMR goats in the compound feed was synthetic (retinol acetate) and has greater bioavailability than the natural vitamins (β-carotene) provided in the pasture for GM and CM goats [29]. Thus, it is logical to hypothesize that TMR kids have higher plasma retinol concentrations than MG and CM kids.

Retinol plasma concentrations in MG and CM kids were higher than those reported by Álvarez et al. [20] in suckling lambs whose mothers were grass fed with a supplementation of oat ad libitum. This result was probably influenced by the vitamin supplementation in the feed of MG and CM goats after parturition [29].

In the current results, we observed that plasma and fat retinol concentrations seem to follow two different metabolic patterns. On the one hand, plasma retinol concentrations depend on the kid requirements of this vitamin, as demonstrated in the dams in the previous experiment of the series [29], as retinol is mobilized from the liver to the plasma to be distributed into target tissues [64]. On the other hand, the dilution effect of the vitamin predominates in fat [44,61]. Therefore, there was not a direct relationship between the concentration of the same compound into the two deposits.

The α-tocopherol plasma concentration was affected by the feeding system (Table 3). The feeding systems with the highest retinol concentration presented the lowest α-tocopherol concentration (MR and TMR), as already indicated. It is possible that the antagonistic effect of vitamin A on the intestinal absorption of vitamin E [61,64] intervened in these results by sharing enzymes or cellular uptake mechanisms [64], as demonstrated in goats [29]. These results reflect the same pattern found in the previous experiment performed on their dams [29]. Thus, this appears to be the only study that shows the same pattern or effect between goats and their kids.

The values of the three natural milk feeding systems (MG, CM, and TMR) were higher than those obtained in the study by Álvarez et al. [20] in suckling lambs whose mothers were grass fed, possibly due to supplementation with vitamin E in our dam feed [29] or in different species.

### 3.4. Discriminant Analysis

Three discriminant analyses (DAs) were carried out using a stepwise model considering the feeding systems of the animals as classification variables. The percentages of correct assignments can be observed in Table 4 for the different cases.

For the first DA, all the parameters related to fat were entered as variables, but only three variables were used for the analysis (% LTF, KFW, and SUM, in this order). In this case, only 67.3% were correctly classified according to their feeding system.

To carry out the second DA, the concentrations of the fat-soluble vitamins in the intermuscular fat were introduced as variables. Three variables (retinol in fat, retinol in plasma, and α-tocopherol in plasma, in this order) were used in this statistical analysis, correctly classifying 72.2% of the kids by cross-validation. The vitamin supplementation of the feed of the goats of all the management systems disallowed better discrimination of the animals. Therefore, these variables have more discriminating power than parameters related to fat (first DA).

Finally, for the third DA, all the parameters studied were entered as variables, again in the order of % LTF, retinol concentrations in the plasma and fat, KFW, SUM, and retinol concentrations in fat were used by the statistical analysis. In total, 94.4% of the kids were classified correctly.

By comparing the three discriminant analyses, we observed that the concentrations of fat-soluble vitamins together with variables related to the fat contents of the kids can be used as traceability tools. This appears to be the only study that proposes traceability tools to classify suckling kids based on the feeding system of the kids and takes into account their dam’s dietary management.

## 4. Conclusions

The vitamin concentrations followed two different concentration patterns in the plasma and fat, with the mobilization and deposition of these compounds according to the requirements of the animal and its metabolism.

The use of vitamins as traceability markers of the production system in suckling kids is not as efficient as in older animals due to the low accumulation of fat in their deposits.

However, the use of these vitamins together with other parameters related to fat allowed adequate discrimination (94.4%) of the animals according to their different feeding systems when considering the different contribution levels of vitamins (on the one hand, natural vs. artificial lactation, and on the other, natural vs. synthetic vitamins).

This study contributes to a better understanding of the traceability of the goat meat production system, as well as to a higher assessment of the quality of this meat, thus contributing to improving the development of this livestock in geographic areas such as the one that has been studied in this paper.

## Figures and Tables

**Figure 1 animals-12-00104-f001:**
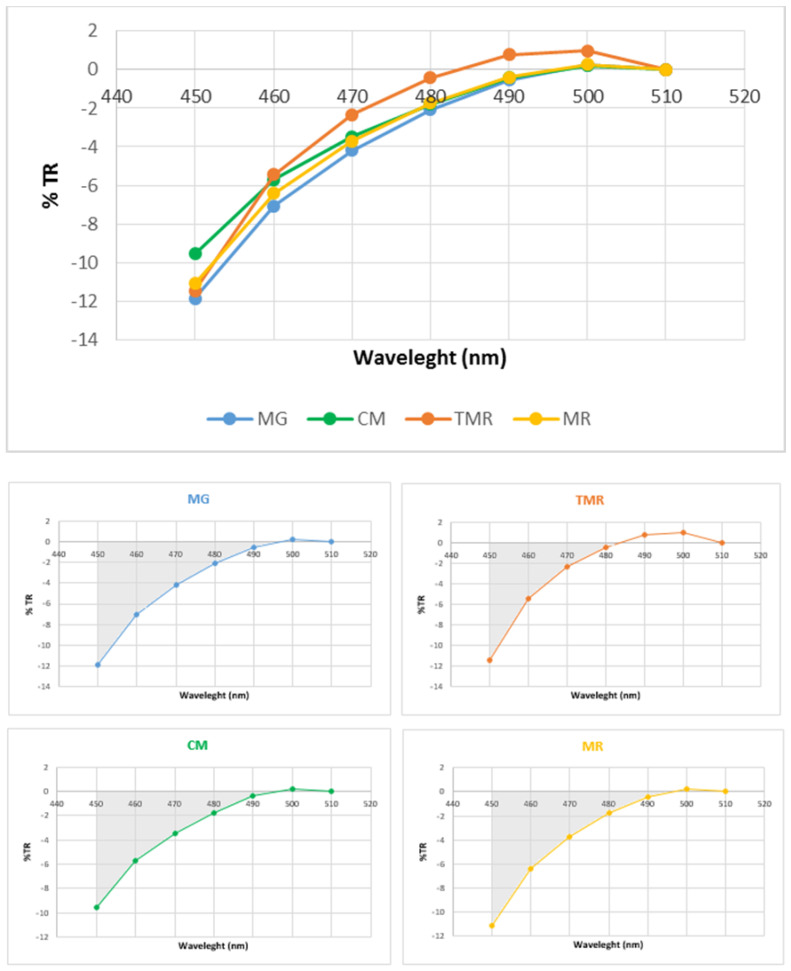
Reflectance spectra pattern of kidney fat for kids reared in four feeding systems. MG: mountain grazing; CM: cultivated meadow; TMR: total mixed ration; MR: milk replacer. The reflectance value (Ri) was translated so that reflectance at 510 nm was equal to zero (TRi). The absolute value of the integral (SUM) is the respective areas comprised between the curve and X-axis in the zone 450–510 nm.

**Table 1 animals-12-00104-t001:** Analytical composition of the milk replacer CORDEVIT CALOSTRADO-50 for milk replacer (MR) kids.

Analytical Components and Additives	Proportions
Crude protein	23.50%
Oils and Fat	26.00%
Crude Fiber	0.10%
Raw ash	7.00%
Calcium	0.90%
Phosphorus	0.70%
Sodium	0.40%
Vitamin A (E-672)	80,000 UI/Kg
Vitamin D3 (E-671)	4,250 UI/Kg
Vitamin E (all-rac-alpha-tocopherol acetate)	30 UI/Kg
Iron (E1) (Sulfate monohydrate)	40 mg/Kg
Iodine (E2) (Calcium iodate anhydrous)	0.15 mg/Kg
Cobalt (E3) (Basic carbonate monohydrate)	0.20 mg/Kg
Copper (E4) (Sulfate Pentahydrate)	5 mg/kg
Magnesium (E5) (Sulfate monohydrate)	25 mg/Kg
Zinc (E6) (Oxide)	30 mg/Kg
Selenium (3b8.12) (Selenomethionine)	0.20 mg/Kg
Antioxidant (B.H.T.) (E-321)	100 mg/Kg
Preservative (Potassium Sorbate) (E-202)	
Emulsifiers (Lecithins) (E-322)	

**Table 2 animals-12-00104-t002:** The mean, standard error, and *p*-value (ANOVA) of the effect of the feeding system on the carcass traits and kid net fat color variables.

	Natural Milk			Milk Replacer	
	MG	CM	TMR	MR	*p*-Value
SA (days)	36.75 ^c^ ± 0.27	31.92 ^b^ ± 0.69	30.07 ^a^ ± 0.44	47.44 ^d^ ± 0.27	<0.001
SLW (kg)	8.53 ± 0.28	8.01 ± 0.22	7.77 ± 0.31	8.38 ± 0.25	0.187
HCW (kg)	4.97 ^b^ ± 0.18	4.30 ^a^ ± 0.15	5.26 ^b^ ± 0.16	4.95 ^b^ ± 0.16	0.002
KFW (g)	53.92 ^a^ ± 3.64	97.85 ^b^ ± 10.99	48.57 ^a^ ± 5.43	37.50 ^a^ ± 5.66	<0.001
	Kidney fat color				
L*	76.79 ± 1.11	73.50 ± 1.49	73.24 ± 1.38	72.77 ± 1.45	0.198
a*	2.83 ^ab^ ± 0.39	1.62 ^a^ ± 0.33	2.24 ^ab^ ± 0.38	2.96 ^b^ ± 0.31	0.036
b*	13.35 ^b^ ± 0.85	11.13 ^a^ ± 0.48	12.96 ^ab^ ± 0.41	13.03 ^ab^ ± 0.48	0.037
C*	13.68 ^b^ ± 0.89	11.30 ^a^ ± 0.50	13.20 ^ab^ ± 0.46	13.39 ^ab^ ± 0.53	0.035
h°	78.44 ± 1.26	82.03 ± 1.41	80.59 ± 1.34	77.57 ± 0.93	0.051
SUM	195.52 ^b^ ±18.05	159.06 ^ab^ ± 14.05	121.40 ^a^ ± 14.27	175.29 ^ab^ ± 12.40	0.007

^a,b,c,d^—Different superscripts in the same column indicate significant differences between feeding systems. MG: mountain grazing; CM: cultivated meadow; TMR: total mixed ration; MR: milk replacer. SA: slaughter age (days); SLW: slaughter live weight (kg); HCW: hot carcasses weight (kg); KFW: kidney fat weight (g) at 24 h post mortem. L*: lightness; a*: redness; b*: yellowness; C*: chroma or saturation; h°: hue angle. SUM: the absolute value of the integral of the translated reflectance spectra.

**Table 3 animals-12-00104-t003:** The mean, standard error, and *p*-value (ANOVA) of the effect of the feeding system on the percentage of intramuscular fat, the lipid oxidation index (mg MDA/kg of meat), and the concentrations of retinol and α-tocopherol in the plasma (μg/mL) and intramuscular fat (μg/g) of the *Longissimus thoracis* muscle in kids reared under four different feeding systems.

	Natural Milk			Milk Replacer	
	MG	CM	TMR	MR	*p*-Value
% LTF	1.13 ^a^ ± 0.13	4.90 ^b^ ± 0.48	4.81 ^b^ ± 0.53	1.63 ^a^ ± 0.21	<0.001
MDA	0.089 ^a^ ± 0.011	0.270 ^b^ ± 0.025	0.252 ^b^ ± 0.039	0.086 ^a^ ± 0.017	<0.001
Fat Retinol	26.81 ^b^ ± 1.30	16.52 ^a^ ± 1.63	15.99 ^a^ ± 1.31	22.63 ^b^ ± 144	<0.001
Plasma Retinol	5.56 ^a^ ± 0.96	3.72 ^a^ ± 0.73	14.98 ^b^ ± 2.748	22.47 ^c^ ± 1.91	<0.001
Plasma Tocopherol	11.43 ^c^ ± 2.50	8.85 ^bc^ ± 2.63	2.49 ^ab^ ± 0.61	0.52 ^a^ ± 0.16	<0.001

^a,b,c^—Different superscripts in the same column indicate significant differences between feeding systems. % LTF: intramuscular fat percentage of the *Longissimus thoracis* muscle. MG: mountain grazing; CM: cultivated meadow; TMR: total mixed ration; and MR: milk replacer.

**Table 4 animals-12-00104-t004:** Discriminant analysis: percentage of animals correctly classified according to their feeding system.

Discriminant	Predicted/Actual Membership	MG	CM	TMR	MR
(1)	MG	66.7	0	0	33.3
	CM	0	69.2	15.4	15.4
	TMR	0	7.1	64.3	28.6
	MR	31.25	0	0	68.75
(2)	MG	83.3	16.7	0	0
	CM	23.1	76.9	0	0
	TMR	0	28.6	42.9	28.6
	MR	0	6.7	6.7	86.7
(3)	MG	100	0	0	0
	CM	7.7	92.3	0	0
	TMR	0	0	92.9	7.1
	MR	0	6.7	0	93.3

(1) According to the fat percentage in L. thoracis (% LTF), kidney fat weight (KFW), and the absolute value of the integral of the translated reflectance spectra (SUM). (2) According to retinol and α-tocopherol concentrations in the plasma and retinol concentration in fat. (3) According to % LTF and retinol and α-tocopherol concentrations in the plasma, KFW, SUM, and retinol concentration in fat. MG: mountain grazing; CM: cultivated meadow; TMR: total mixed ration; MR: milk replacer.

## Data Availability

Not applicable.

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
