# Peer review of "Retinol and α-Tocopherol Contents, Fat Color, and Lipid Oxidation as Traceability Tools of the Feeding System in Suckling Payoya Kids"

_animals, 2022, doi:10.3390/ani12010104_

Round 1

Reviewer 1 Report

This manuscript is generally well structured and written,  I don't have any concerns about structure, content or statistical analysis approach. Must be accept in present form.

Author Response

Thanks for your consideration.

Reviewer 2 Report

General comments/questions

After reviewing this paper some questions arose, please consider these

questions:

As is said the young age of the animals limited fat deposition and the cumulative effect of carotenoids and fat-soluble vitamins, so this measurements are relevant in so young kids goats?

Why to evaluate lipid oxidation or retinol and α-tocopherol contents in so young animals?

What are the implications of this study? Is it supposed to apply different meat prices according to animal feeding?

How do you incorporate this information in meat market?

What is the importance and the pratical implications of this study?

These questions should be addressed througout the article, particularly in introduction and conclusion.

Specific comments:

Line 146: Please give details of the analytical composition of the commercial feed.

Line 163: How do you justify this difference in slaughter age?

Line 171: kids were only weighed at slaughterhouse?

Line 182: Specify the precise site of longissimus thoracis excision

Lines 190-191: It means that kidney fat color was measured differently? Please explain and specify.

Line 209: Delete full stop

Line 222-223: Delete (1 g…. ) curved brackets

Line 293: Usually superscript a is used for the higher value, please verify. It is confusing.

Line 322-323: It’s incorrect. Article 18 refers to light lambs not goat kids. Please verify. If possible, use references of goats not lambs. Be carefull with these comparisons.

Line 392 - Table 3: The values of %LTF are in agreement with other studies? Please discuss.

Line 410: Have you quantified the fatty acid profile of CM and TMR diets? Are they rich in linoleic acid and linolenic acid? Justify and complete.

Line 440-447: Again, why to evaluate lipid oxidation or retinol and α-tocopherol contentes in intramuscular fat in so young animals?

Line 523: Add the limitations and pratical implications of this study, and how this evaluation could be improved for better results and accuracy.

Reviewer 4 Report

Introduction

Lines-56-69: I think the information provided ins interesting. However, the information is useless for the reader of the manuscript, unless they are from the same region of the authors. I understand that this contextualization is important for the authors for several reasons, but I don`t think it adds much to the paper in an international journal. With that being said, I would suggest to the authors to withdraw the first and second paragraphs of the manuscript.

Material and methods

Line 177 (and at other places such as Bligh and Dyer description method): Please add the centrifugation speed as x g and not RPM;

Lines 210-221: I am afraid that the method used (Bligh and dyer method) is not an accurate method for fat quantification. It has been really debated in the field the data obtained by fat extraction through this method as a way to quantify the amount of fat in the sample. For my understanding, this is not an appropriate method to do so because it lacks reproducibility, and it has also a lot of bias on the step of separation of the sample from the extract (lipids+chloroform-methanol). I understand that it should be used to obtain fat samples for quality analysis but not for quantification, which must be done by other appropriate methods with higher accuracy (i.e. AOCS, 2009 procedure Am 5-04).

Lines 274-284: I would suggest to the authors to insert the equation of their statistical model to facilitate the understanding of the statistical analysis by the readers.

General question: Have the authors considered to measure the amount of milk produced by the dam in their model? It was not clear for me if the amount of milk that was taken by sucking kids was the same for the ones that was receiving milk replacement. If there is any chance that the kids receiving milk replacer were fed less than the other groups, it may explain the results of slaughter age and likely the KFW. As such, if there is any chance to have the amount of milk suckled by the kids from the other groups, it would help to clear the results (by using it as a co-variate). As it is now, the main question is: does the results observed are due to the difference in composition/quality of milk x replacer or because of the amount the kids received? I think it is something worth to think about.

Results and Discussion

Line 337: “detect an effect”

Lines 377-378: Very arguably result about fat content by measuring it by the method used. I would not include fat content as a variable in the study as I consider the method used not eligible to get accurate results. That`s one decision the authors (and may be the editor) will take. This is only a suggestion as I know a lot of papers already published have used it before.

Conclusions

Lines 532-533: It does need to be state in your conclusion section. I would just delete it.  

Round 2

Reviewer 2 Report

Lines 539-545 should be in introduction after lines 116-117.

In conclusions add a short statement on the contribution of this particular study.

Author Response

Based on the reviewer's suggestions. The introduction and conclusions have been improved. 

Reviewer 4 Report

The authors have addressed the questions and changed the manuscript taking into consideration the comments made.

I had two major concerns that was addressed by the authors. One of them regards to the quantitative measurement of fat by B&D method. I understand that there is now way to going back to the study now and I see no issue to move forward to with the publication of the manuscript. However, it must be emphasized that, despite the fact that previous papers have been published by doing the same method, there was a huge evolution on fat analysis and it is well known that B&D is not as accurate as other methods, so, science needs to move on. However, I understand that in this case, particularly, it won`t be an issue since all treatments had their fat content measured by the same method. So, any bias would affected both treatments  similarly.

Regarding the other comment made, about the amount of milk produced, i also understand that since the authors do not have the data, thee is nothing we can do about it at this point. It would be great if we could see it. Even though it was not expected to be different between treatments, to have in in the experimental model as a known effect would make the results even clearer, and thats why I have asked and suggested its inclusion in my last review.  However, I agree with the authors that despite the lack of this information, the manuscript is worth to be published. 

Author Response

We are very grateful for all the suggestions that the reviewer has made, with which we agree. We take note of all this, and we will try to implement them in future studies.